# Synthesis and Investigation of the Analgesic Potential of Enantiomerically Pure Schiff Bases: A Mechanistic Approach

**DOI:** 10.3390/molecules27165206

**Published:** 2022-08-15

**Authors:** Hamid Hussain Afridi, Muhammad Shoaib, Fakhria A. Al-Joufi, Syed Wadood Ali Shah, Haya Hussain, Abid Ullah, Mohammad Zahoor, Ehsan Ullah Mughal

**Affiliations:** 1Department of Pharmacy, University of Malakand Dir (Lower) at Chakdara, Chakdara 18800, KPK, Pakistan; 2Department of Pharmacy, Shaheed Benazir Bhutto University Sheringal Dir (Upper), Dir 18000, KPK, Pakistan; 3Department of Pharmacology, College of Pharmacy, Jouf University, Aljouf 72341, Saudi Arabia; 4Department of Chemistry, University of Malakand Dir (Lower) at Chakdara, Chakdara 18800, KPK, Pakistan; 5Department of Chemistry, University of Gujrat, Gujrat 50700, PB, Pakistan

**Keywords:** Schiff bases, analgesic activity, acetic-acid-induced writhing assay, tail immersion assay, hot plate assay, naloxone, opioid receptor

## Abstract

Schiff bases are a class of organic compounds with azomethine moiety, exhibiting a wide range of biological potentials. In this research, six chiral Schiff bases, three ‘S’ series (**H1**–**H3**) and three ‘R’ series (**H4**–**H6**), were synthesized. The reaction was neat, which means without a solvent, and occurred at room temperature with a high product yield. The synthesized compounds were evaluated for analgesic potential in vivo at doses of 12.5 and 25 mg/kg using acetic-acid-induced writhing assay, formalin test, tail immersion and hot plate models, followed by investigating the possible involvement of opioid receptors. The compounds **H2** and **H3** significantly (*** *p* < 0.001) reduced the writhing frequency, and **H3** and **H5** significantly (*** *p* < 0.001) reduced pain in both phases of the formalin test. The compounds **H2** and **H5** significantly (*** *p* < 0.001) increased latency at 90 min in tail immersion, while **H2** significantly (*** *p* < 0.001) increased latency at 90 min in the hot plate test. The ‘S’ series Schiff bases, **H1**–**H3**, were found more potent than the ‘R’ series compounds, **H4**–**H6**. The possible involvement of opioid receptors was also surveyed utilizing naloxone in tail immersion and hot plate models, investigating the involvement of opioid receptors. The synthesized compounds could be used as alternative analgesic agents subjected to further evaluation in other animal models to confirm the observed biological potential.

## 1. Introduction

The term pain derives from the Latin and the Greek peone and poine, respectively, meaning “penalty” or “punishment” [1]. Pain is an “emotion” and the unpleasant sensory experience that is linked with potential or actual tissue damage [2]. The North American Nursing Diagnosis Association has defined pain as a condition in which the patient feels and reports severe discomforts and uncomfortable sensation (pain reporting may be descriptor-encoded or verbal communication) [3]. A type of pain that usually abates rapidly is called acute pain. This type of pain is physiological and provides a warning sign about the disease and the unsafe conditions of a person as to something bad that has occurred in the body; mostly, it is nociceptive in nature but can be neuropathic too, and most associated etiologies are trauma, surgery, labor, acute illness and some special procedures [4].

A type of pain that exists for a longer period (from months to years) is called chronic pain and is different from acute pain in that it can be neuropathic, nociceptive or both [5]. Its further subtypes are: (i) pain that is associated with chronic disease (from secondary pain to osteoarthritis) and (ii) pain lacking a recognizable organic cause (e.g., fibromyalgia) [6]. Pain associated with the most severe situations, that is, cancer or malignancy, is simply named as cancer or malignant pain. This type of pain comprises both the acute and chronic types and have several causes ranging from the disease itself to the use of diagnostic tools (incisions for biopsy) and treatment (radiation and chemotherapy) [7]. Other types include nociceptive pain, which is further categorized into somatic and visceral pain. Somatic pain is concerned usually with the external body and originates from joints, bones, skin, muscles and connective tissues, while visceral pain mostly originates from internal organs, means occurring secondary to hepatic metastasis, pancreas, gall bladder and small intestine obstruction [8]. Neuropathic pain is usually due to pressure or injury of the nerves, leading to nerve damage. This type of pain is chronic in nature and does not heal and subside early as in the case of acute pain [9].

Schiff bases are organic compounds with imine or azomethine (–C=N–) functional groups and are the condensation products of primary amines with carbonyl compounds formed under acid–base catalysis [10]. Previously reported asymmetric Schiff bases were synthesized in the presence of organic solvents; in this paper, we used an environmentally friendly solvent-free methodology for the synthesis of chiral Schiff bases. Schiff bases are symmetrical and some are asymmetrical, such as sulfonamide derivatives of Schiff bases that act as asymmetrical catalysts [11]. Schiff bases are of key importance as they have various biological activities and are effective against pathogens [12]. It has been reported that chiral Schiff bases exhibit tremendous antimicrobial activities [13]. Schiff bases are used for pharmacological purposes, such as the treatment of cancer, and have a pivotal role in optical [14], medical, analytical and bioorganic applications [15]. Some of the Schiff bases had better activity against *Plasmodium falciparum strain* (ACC) [16], whereas some have shown anti-cancer [17] and anti-fungal activities [18].

Keeping in view the importance and applications of Schiff bases, the current study aims to prepare chiral Schiff bases and screens them for their analgesic potential.

## 2. Results

### 2.1. Synthesis and Characterization of the Schiff Bases

A series of chiral Schiff bases, three ‘S’ series **H1–H3** (Figure 1) and three ‘R’ series **H4–H6** (Figure 2), on aromatic rings having substitution with several functional groups were synthesized in this study by treating commercially available aldehydes with chiral amines after being ground in mortar and pestle for a period of 5–10 min at room temperature without the addition of any solvent or catalyst. The product resulted in a paste and was left overnight to dry and finally recrystallized in methanol (Figure 1 and Figure 2).

### 2.2. Pharmacological Activities

#### 2.2.1. Acetic-Acid-Induced Writhing Method

The analgesic potential of the compounds **H1**–**H6** was investigated by administering the doses of 12.5 and 25 mg/kg body weight into mice and its effect on pain was evaluated using the acetic-acid-induced writhing model. Writhing is a specific action in which the mice extend its abdomen and stretch hind limbs after injecting acetic acid to mice. The frequency of writhing was observed and calculated where compound **H2** exhibited a significant decrease in the total writhes with the tested doses. At doses 12.5 and 25 mg/kg body weight, **H1** showed 68.31% (17.87 ± 2.09, *p* < 0.01) and 75.08% (14.05 ± 2.11, *p* < 0.001) inhibitory effects, respectively, in comparison to that of the control group (56.39 ± 1.66), as presented in Figure 1. The Schiff bases were also effective at the tested doses. Among them, the most promising effects were observed for **H3**, which were found to be 68.49% (17.77 ± 1.89, *p* < 0.001) and 72.99% (15.23 ± 2.03, *p* < 0.001), corresponding to doses 12.5 and 25 mg/kg body weight (b.w). The standard diclofenac sodium at a dose of 10 mg/kg b.w showed 84.55% (8.71 ± 1.51) activity.

The ‘S’ series chiral Schiff bases (**H1–H3**) showed significantly reduction in the number of writhes. It was a similar case for the ‘R’ series Schiff bases (**H4–H6**), in which **H5** exhibited a maximum effect, 70.12% (16.85 ± 2.05, *n* = 6) at 25 mg/kg b.w, followed by **H4** at 68.47% (17.78 ± 1.36) at the same dose tested. The responses of ‘R’ series were comparable to those of the ‘S’ series compounds.

#### 2.2.2. Formalin-Induced Paw-Licking Time

Pain induced by the formalin assay was also employed to investigate the analgesic effect of **H1**–**H6** at the doses of 12.5 and 25 mg/kg b.w. The **H1** exhibited the highest analgesic potential and lowered the pain by 49.40% (17.77 ± 1.07, *p* < 0.01) and 59.94% (14.07 ± 2.21, *p* < 0.001), correspondingly to the tested afore-mentioned doses in first phase, and 65.92% (20.95 ± 2.21, *p* < 0.05) and 73.95% (16.01 ± 1.42, *p* < 0.001) in the second phase, as compared to the control group (35.12 ± 1.22 and 61.47 ± 1.39), which is shown in Table 1.

Similarly, **H2**, at a dose of 25 mg/kg b.w, showed promising analgesic potential, where the decrease in the pain was 69.48% (10.72 ± 1.35, *p* < 0.001) and 78.53% (13.2 ± 2.05, *p* < 0.001), in first and second phases, respectively, as compared to the control group’s results (35.12 ± 1.22 and 61.47 ± 1.39, respectively).

The Schiff base **H3** showed promising effects at both doses: 57.94% (14.77 ± 1.49, *p* < 0.001, *n* = 6) and 58.06% (14.73 ± 2.26, *p* < 0.001) in phase 1, whereas 64.58% (21.77 ± 1.89, *p* < 0.05) and 71.97% (17.23 ± 1.76, *p* < 0.01) in phase 2, as compare to the control group (35.12 ± 1.22 and 61.47 ± 1.39), while **H5** showed 61.25% (13.61 ± 2.23, *p* < 0.001) in phase 1 and 71.48% (17.53 ± 1.15, *p* < 0.001) analgesic potential in phase 2 at a dose of 25 mg/kg b.w.

Animals pretreated with indomethacin (10 mg/kg) showed a considerable decrease in the paw-licking response of 76.10% (9.47 ± 1.37, *p* < 0.001) in the second phase, whereas a mild decline was found in the paw-licking time at 22.30% in the first phase.

Current investigation showed that the tested doses of **H1**–**H6** shortened the frequency of paw-licking response during inflammation. Indomethacin, being a standard at a dose of 10 mg/kg b.w, produced 84.55% (26.48 ± 1.33, *p* < 0.05) and 78.50% (13.21 ± 1.39, *p* < 0.001) of the effects.

#### 2.2.3. Tail Immersion Method and Possible Involvement of Opioidergic System

The tail immersion model results of **H1**–**H6** are shown in Table 2. The maximum analgesic effect was produced by **H2**, which was 65.96% (3.76 ± 0.049, *p* < 0.001) recorded after 90 min, whereas the latency period increased at the dose of 25 mg/kg b.w. The latency response was counted in seconds, where **H3** expressed a significant upsurge in latency at the dose of 12.5 mg/kg b.w and 53.11% (2.73 ± 0.035, *p* < 0.01) and 59.37% (3.15 ± 0.028, *p* < 0.001, *n* = 6) at 25 mg/kg b.w, in comparison to the control group (1.28 ± 0.023).

In addition to these two, other Schiff bases were also equally effective. Among them, the most promising effects were observed for **H5**, 43.86% (2.28 ± 0.036, *p* < 0.001) and 59.49% (3.16 ± 0.052, *p* < 0.001), respectively, for the tested doses in comparison to the control group (1.28 ± 0.023). Tramadol is a standard opioid drug that exhibited a noteworthy post-treatment potency that was found to be 84.43% (8.22 ± 0.034, *p* < 0.001), when compared with the control group (1.28 ± 0.023). The other standard, morphine (opioid analgesic drug), provided substantial post-treatment potency, which was found to be 85.23% (8.67 ± 0.029, *p* < 0.001).

Animals pretreated with naloxone substantially antagonized the analgesic potential of morphine and tramadol (centrally acting opioid analgesics), indicating the involvement of the opiodergic system in an analgesic response. However, pretreatment with naloxone causes partial decline in the analgesic response of **H1**–**H6** in a dose-dependent manner in mice, indicating the partial involvement of opioid receptors (Table 3).

#### 2.2.4. Hot Plate Test for the Assessment of Analgesic Activity

The analgesic response of the synthesized Schiff bases (H1–H6) using the hot plat test was also estimated, as presented in Table 4. The extreme analgesic effect of **H2** was observed at 90 min where latency was increased. The latency response counted in seconds was high for **H2**, which expressed a significant increase in latency at a dose of 12.5 and 25 mg/kg b.w, 61.32% (4.24 ± 0.039, *p* < 0.01) and 70.56% (5.57 ± 0.048, *p* < 0.001) correspondingly. For the control group, the recorded response was 1.28 ± 0.023.

The Schiff base **H1** at a dose of 12.5 mg/kg b.w showed a 61.14% (4.22 ± 0.038, *p* < 0.01) response, whereas 69.40% (5.36 ± 0.046, *p* < 0.001) at the second tested dose of 25 mg/kg b.w, which was comparable to the control group’s response recorded as 1.28 ± 0.023. Similarly, **H4** at a dose of 25 mg/kg b.w showed a response of 64.04% (4.56 ± 0.052, *p* < 0.001) in comparison to the control group’s response, 1.28 ± 0.023, while **H5** showed a 64.50% (4.62 ± 0.045, *p* < 0.001) response at the afore-mentioned dose.

Tramadol produced a noteworthy post-treatment potency, which was recorded as 83.79% (10.12 ± 0.038, *p* < 0.001, *n* = 6) after compared with the control group’s response of 1.28 ± 0.023. Morphine also produced a high post-treatment potency response, which was found to be 84.68% (10.71 ± 0.022, *p* < 0.001) in comparison to the control group’s response of 1.28 ± 0.023.

Animals pretreated with naloxone substantially antagonized the analgesic potential of morphine and tramadol (centrally acting opioid analgesics), indicating the involvement of the opiodergic system in analgesic response. However, pretreatment with naloxone causes a partial decline in the analgesic response of **H1**–**H6** in a dose-dependent manner in mice, indicating the partial involvement of opioid receptors (Table 5).

## 3. Discussion

In this study, we synthesized six chiral Schiff bases belonging to the *‘S’* series (**H1–H3**) and *‘R’* series (**H4–H6**) in high yield using grindstone chemistry, which is an environmentally friendly procedure involving no catalysts or solvents.

Various pain models are used to evaluate analgesic potentials of natural and synthetic compounds. The most important and frequent in use among them are acetic-acid-induced writhing, the formalin pain test, tail flick test and the hot plate test in mice or rats [19]. Pain mechanism involve three main events due to noxious stimulation: transmission, transduction and modulation of the signals [20]. The writhing test has been suggested for the preliminary measurement of anti-nociceptive effects on potential substances [21]. In this test, nociceptors activation followed by an expected inflammation of the viscera are carried out by chemically noxious substances, such as glacial acetic acid (0.3–0.6%), 2-phenyl-1,4-benzoquinone and magnesium sulfate, which are administered to rodents (frequently mice) by the peritoneal cavity route [22]. Acetic acid causes the induction of pain in mice due to its localized inflammatory response of free arachidonic acid released from tissue phospholipids through COX-producing prostaglandin [23]. The post-injection period up to an hour includes writhing reactions such as stretching, twisting, extension of hind legs and contraction of the abdomen [24,25]. The acetic-acid-induced writhing test is suggested for peripherally acting drugs (antihistamine, meprobamate and chlorpromazine) and is thus employed as the preferred analgesic assay [26]. In this paper, all the tested compounds exhibited analgesic potential; however, among them, H2 from the ‘S’ series and **H5** from the ‘R’ series were comparatively more potent. These compounds thus have the potential to be used as alternative analgesic drugs in the future, subjected to toxicological evaluation in different animal models.

Among the assays, the formalin test is considered as a satisfactory model to evaluate the analgesic drugs for their efficacies. This test is executed for acute and long lasting pain evaluation in the paws of mice or rats [22], which are sensitive towards centrally acting analgesic drugs [27]. The three main behavioral responses carried out by the formalin injection are phasic flexion, tonic flexion and licking of the injected limb after injecting intraplantar formalin in the paw of mice or rats [28]. The direct stimulation of nociceptors occurs in the first phase, such as C-fibre and low-threshold mechanoreceptors, including the upregulation of substance P; then, inflammatory phenomena occur involving the central sensitization of rodents within the dorsal horn neurons, including the upregulation of histamine, serotonin, bradykinin and prostaglandin [29]. A similar trend in efficacy was observed here also and, as representative compounds of each series, **H2** and **H5** were more potent than the other tested compounds.

For the evaluation of the centrally acting drugs, the tail immersion and hot plate methods are equally used and, in these tests, the nociception pathway is mostly involved. Tail emersion in hot water produces spinally mediated nociception, whereas the hot plate method is only selective for the supraspinally mediated nociception [30,31]. Thus, the initiation of the analgesic action of opioid agents involves the spinal and supraspinal receptors [32]. Our results show that the compounds **H1**–**H6** inhibited nociception, which are associated with spinal and supraspinal opioid receptors in the central nervous system. The anti-nociceptive effect of compounds **H1**–**H6** verified this hypothesis by naloxone antagonism and it was partially produced in the hot plate and tail flick models, whereas naloxone is a nonselective opioid receptor antagonist [33]. In both experimental models, naloxone reduced the morphine-induced latency time at the given concentration. The current findings suggest that the activity of the tramadol (standard opioid analgesic) is parallel with that of the tested samples, indicating the potential of these compounds to be used as analgesic drugs. In the same way, the analgesic action of the tested samples and tramadol was considerably exaggerated by the earlier administration of naloxone, thus expressing the involvement of the opioid receptors in their modes of action.

Animals pretreated with naloxone substantially antagonised the analgesic potential of morphine and tramadol (centrally acting opioid analgesics), indicating the involvement of the opiodergic system in analgesic response. However, pretreatment with naloxone causes a partial decline in the analgesic response of **H1**–**H6** in a dose-dependent manner in mice using the tail immersion and hot plate methods, indicating the partial involvement of opioid receptors and other mechanisms, which requires further investigation to confirm the mechanisms involved in the management of pain observed in this study.

Taken all together, this current study testified the role of chiral Schiff bases (**H1–H6)** as analgesic agents, which may act as promising candidates, and this warrants particular consideration in the research and development of the management of pain.

## 4. Materials and Methods

### 4.1. Materials

All chemicals and solvents used in this work were of analytical grade, manufactured by Sigma Aldrich (Merck), Germany, and were obtained from the local market. The progress of the reaction was monitored by TLC (Thin-layer chromatography) on Merck 60F_254_ silica-gel-coated plates. The structures of the synthesized compounds were confirmed by ^1^H-NMR spectra (300 MHz) on Bruker Varian Mercury 300 MHz FT Spectrometer, in CDCl_3_, FT-IR.

### 4.2. Methodology

We synthesized six chiral Schiff bases (**H1**–**H6**) among which **H1–H3** were ‘S’ series compounds and **H4–H6** were ‘R’ enantiomers prepared via a solvent-free grinding method.

### 4.3. General Procedure for the Synthesis and Characterization of Chiral Schiff Bases

Chiral Schiff bases were synthesized by treating commercially available aldehydes with chiral amine by grinding in a mortar and pestle for a period of 5–10 min at room temperature without the addition of any solvent or catalyst. The reaction progress was monitored by TLC. A paste was obtained upon grinding. After the completion of the reaction, water from the mixture was evaporated until dryness. The powder product obtained was finally recrystallized in absolute methanol [34]. The synthesized compounds were characterized through different spectroscopic techniques, including Proton Nuclear magnetic resonance spectroscopy (^1^H NMR) and infrared Spectroscopy. The ^1^H NMR was carried out in deuterated chloroform (CDCl3) on Bruker Avance II 400 MHz NMR Spectrometer using 300 MHz frequency for all compounds, except compound H1 which was run on 400 MHz frequency. The spectra were resolved through Topspin-4 (Bruker, London, UK) software. The IR spectroscopy was performed on Spectrum 3™ FT-IR Spectrometer.

As mentioned, all the synthesized compounds were successfully characterized through H NMR and IR Spectroscopic techniques. The technical details are described as follows:

#### 4.3.1. Synthesis of (S,E)-N,N-dimethyl-4-(((1-phenylethyl)imino)methyl)aniline (H1)

Yield: 78.5%, pale yellowish crystals, M.P: 70–76 °C, solubility: chloroform, ethanol, Rf value: 0.57. ^1^H NMR (CDCl_3_, 400 MHz): δ 8.28 (s, 1H), 7.79–7.76 (m, 1H), 7.71 (d, *J* =12.0 Hz, 2H), 7.45–7.48 (m, 2H), 7.36 (t, *J* =9.6 Hz, 2H), 7.25–7.28 (m, 1H), 6.71 (d, *J* =12.0 Hz, 2H), 4.51 (q, 1H), 3.03 (s, 6H), 1.61 (d, *J* =8.8 Hz, 3H) (Appendix A). IR (KBr, cm^−1^, Appendix A): 2985.71, 2823.79, 1627.92, 1602.85, 1525.69, 1365.60, 1185.1, 1093.64 REF. Reported by Kang et al. [35].

#### 4.3.2. Synthesis of (S,E)-N-(4-methylbenzylidene)-1-phenylethanamine (H2)

Yield: 88.4%, white crystals, Solubility: chloroform, ethanol, M.P: 105–109 °C, Rf: 0.73. ^1^H NMR (300 MHz, CDCl_3_): δ 8.37 (s, 1H), 7.70 (d, *J* = 8.0 Hz, 2H), 7.23–7.8 (m, 9H), 4.57 (q, 1H), 2.41 (s, 3H), 1.64 (d, *J* =7.2 Hz, 3H) (Appendix A). IR (KBr, cm^−1^, Appendix A): 2957.25, 2825.12, 1637.56, 1487.12, 1377.17, 1087.85 [36].

#### 4.3.3. Synthesis of (S,E)-N-(4-nitrobenzylidene)-1-phenylethanamine (H3)

Yield: 73.6%, brown crystals. solubility: chloroform, ethanol, M.P: 98–102 °C, Rf: 0.63. ^1^H NMR (300 MHz, CDCl_3_): δ 8.22–8.41 (m, 5H), 7.16–7.37 (m, 4H), 4.58 (q, 1H), 1.59 (d, 3H) (Appendix A). IR (KBr, cm^−1^, Appendix A): 3080.30, 3000.18, 1645.41, 1466.12, 1310.17, 1057.85 [36].

#### 4.3.4. Synthesis of (R,E)-N,N-dimethyl-4-(((1-phenylethyl)imino)methyl)aniline (H4)

Yield: 80%, pale yellow crystals, solubility: chloroform, ethanol, M.P: 77–81 °C, Rf: 0.55. ^1^H NMR (300 MHz, CDCl_3_): δ 8.27 (s, 1H), 7.67 (d, *J* = 8.8 Hz, 2H), 7.44 (d, *J* = 7.3 Hz, 2H), 7.34 (t, *J* = 7.5 Hz, 2H), 7.25 (d, *J* = 7.1 Hz, 1H), 6.71 (d, *J* = 8.8 Hz, 2H), 4.50 (q, *J* = 6.5 Hz, 1H), 3.03 (s, 6H), 1.60 (d, *J* = 6.6 Hz, 3H) (Appendix A). IR (KBr, cm^−1^, Appendix A): 2985.71, 2823.79, 1627.92, 1602.85, 1525.69, 1365.60, 1185.1, 1093.64. Kang, Lee, Lee, Lee, Lee, Cho and Hur [35].

#### 4.3.5. Synthesis of (R,E)-N-(4-methylbenzylidene)-1-phenylethanamine (H5)

Yield: 86%, white crystals, solubility: chloroform, ethanol, M.P: 108–112 °C, Rf: 0.73. ^1^H NMR (300 MHz, CDCl_3_): δ 8.37 (s, 1H), 7.71 (d, *J* = 8.0 Hz, 2H), 7.40 (dt, *J* = 13.0, 7.4 Hz, 4H), 7.30–7.20 (m, 3H), 4.56 (q, *J* = 6.6 Hz, 1H), 2.41 (d, 3H), 1.62 (s, *J* = 6.6 Hz, 3H) (Appendix A). IR (KBr, cm^−1^, Appendix A): 2957.25, 2825.12, 1637.56, 1487.12, 1377.17, 1087.85 [36].

#### 4.3.6. Synthesis of (R,E)-N-(4-nitrobenzylidene)-1-phenylethanamine (H6)

Yield: 74%, brown crystals, solubility: chloroform, ethanol, M.P: 97–101 °C, Rf: 0.65. ^1^H NMR (300 MHz, CDCl_3_): δ 8.41 (m, 3H), 7.92 (m, 2H), 7.35 (m, 4H), 4.57 (q, 1H), 1.49 (d, 3H) (Appendix A). IR (KBr, cm^−1^, Appendix A): 3060.30, 2970.18, 1635.41, 1453.12, 1208.17, 1033.45 [36].

### 4.4. Pharmacological Activities

#### 4.4.1. Animals, Dosing and Grouping

Balb/C mice of weight 18–25 g were procured from National Institute of Health, Islamabad, and kept in the animal house of University of Malakand Dir (Lower) at Chakdara. The animals were kept in a dark and light cycle of about 12 h each (light at 6:00 am) and having relative humidity in the range of 50–55% at room temperature at about 22–25 °C. A study was conducted as per the approval from the Departmental Ethical Committee (Pharm/EC/75-01/20), (Scientific Procedures Issue-I of the University of Malakand) in compliance with provisions of the Animal Bye-Laws 2008.

#### 4.4.2. Acute Toxicity Study

The synthesized Schiff bases (H1–H6) were evaluated for optimum dose determination for the in vivo studies. In two stages, different groups of animals were administered synthesized compounds orally. In stage I, animals received 6.25, 12.5, 30, 50, 75, 100 and 125 mg/kg doses, and 25, 50, 75, 100, 150, 200 and 250 mg/kg doses in stage II. The animals were observed continuously for 24 h after the dosing for toxicological symptoms such as lacrimation, tremors, convulsions, salivation, and loss of righting reflex, motor activity, sedation, hypnosis, muscle spasm and diarrhea. Animals were further observed for mortality up to 72 h and the tested compounds were found non-toxic in stage I and safe up to 125 mg/kg and, in stage II, were safe and non-toxic up to 250 mg/kg dose. Therefore, according to the OECD guidelines, the 12.5 mg/kg dose, which was 1/10th of 125 mg/kg, and 25 mg/kg dose, which was 1/10th of 250 mg/kg treated dose, were selected the suitable doses for further in vivo studies [36].

#### 4.4.3. Acetic-Acid-Writhing Test

The analgesic potency of the synthesized chiral Schiff bases was carried out by the acetic-acid-induced writhing test. Mice were categorized into different groups, having 6 mice in each group. The control group received 2% Tween 80, and the tested compounds were administered at the 12.5 mg/kg and 25 mg/k p.o doses. Diclofenac sodium 10 mg/kg i.p served as the positive control. Acetic acid 0.1 mL (0.6%) was injected *i.p* to mice after a 15 min interval. The number of the writhes performed in each of the mouse was observed and noted for about 30 min by placing on a flat surface after the administration of the acetic acid [37]. The percentage inhibition of writhing was determined by the following formula:(1)$$\mathrm{Percent}\;\mathrm{Inhibition}=\frac{\mathrm{No}.\mathrm{of}\;\mathrm{control}\;\mathrm{group}\;\mathrm{writhes}\;–\;\mathrm{No}.\mathrm{of}\;\mathrm{test}\;\mathrm{group}\;\mathrm{writhes}}{\mathrm{No}.\mathrm{of}\;\mathrm{control}\;\mathrm{group}\;\mathrm{writhes}\;\times 100}$$

#### 4.4.4. Formalin Test

The formalin test was carried out in animals to assess the analgesic activity of the synthesized Schiff bases according to the standard procedure. Animals were randomly divided in different groups each having 6 mice and pre-treated (30 min) with normal saline (2% Tween 80); the synthesized compounds and indomethacin 10 mg/kg (standard drug) were injected intraperitoneally (i.p.) 15 min before the administration of formalin. The experimental groups received doses of 12.5 and 25 mg (p.o) per kg body weight of the synthesized compounds followed by the sub-cutaneous injection of formalin at 20 μL (2.5%) into the dorsal surface of the right hind paw of each animal [38]. The time of pain response (paw-licking time) for each animal group was observed and calculated in two periods, i.e., 0–5 min (direct activation of nociceptors) and 15–45 min (inflammatory-mediators-induced pain response), and compared with the control group animals [39].

#### 4.4.5. Tail Immersion Test

The mice were used for checking the central analgesic potential of the synthesized Schiff bases. Mice were divided in different groups and the synthesized compounds at the doses of 12.5 and 25 mg as well as the normal saline (2% Tween 80) solution were administered to the animals at 30 min and morphine 2 mg/kg (standard; *i.p*) at 15 min before the test to the respective groups. The latency period (time of reaction measured) at 15, 30, 45, 60, 75 and 90 min after the administration of the test samples was recorded using a stopwatch (time measured for tail deflection by mice) and was calculated by immersing the tail of mice (1–2 cm) in hot water (temperature maintained at 55 ± 1 °C) [40].

#### 4.4.6. Hot Plate Test

The Eddy and Leimbach method was used for assessing the analgesic activity of the synthesized Schiff bases with slight modifications in the hot plate model. Animals were divided in different groups each carrying 6 and placed on a hot plate at a temperature of 54 ± 2 °C and the time taken by animal for either jumping or paw licking were recorded. The latency time until animal discomfort (jumping, hind paw licking or hind paw lifting) 60 s cutoff time was recorded at different time intervals after the administration of normal saline (2% Tween 80 solution), synthetic compounds (12.5 and 25 mg/kg), tramadol (10 mg/kg) and morphine (2 mg/kg) [41].

#### 4.4.7. Molecular-Level Mechanisms for Opioid Receptor Involvement

After the acetic-acid-induced writhing model, formalin test, tail emersion test and hot plate method, the synthesized compounds were subjected to mechanistic approach for the possible association of opioid receptors. In order to confirm the connection of opioid receptors, a previously reported procedure was followed [32]. In this assay, different groups of mice, each having 6 individuals, were pre-treated with an antagonist of non-selective opioid receptor; naloxone 2 mg/kg (*i.p*) was injected 30 min prior to the given dose of the pure compounds (12.5 mg/kg and 25 mg/kg *p.o*) and morphine (2 mg i.p), and were tested by using the tail immersion and hot plate tests, respectively.

## 5. Conclusions

In this research work, six chiral Schiff bases, three of ‘S’ series (**H1–H3**) and three of ‘R’ series (**H4–H6**), were synthesized and evaluated for analgesic potential in mice models. The synthesized Schiff bases showed promising analgesic responses in all the tested models in reducing the induced pain threshold, which suggests their partial involvement of the opioidergic system in their mode of actions. Thus, the synthesized compounds might be promising candidates that warrant particular consideration in the research and development of the management of pain. Further studies are needed to explore the exact mechanisms involved in the management of pain as these are preliminary results.

## Data Availability

Not applicable.

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
