# Peer review of "Synthesis and Investigation of the Analgesic Potential of Enantiomerically Pure Schiff Bases: A Mechanistic Approach"

_molecules, 2022, doi:10.3390/molecules27165206_

Round 1
Reviewer 1 Report
In this manuscript, the authors reported six chiral Schiff bases H1—H3/H4-H6 and these compounds evaluated for analgesic potential at a dose 12.5/25 mg/kg using AcOH induced writhing, formalin test, tail immersion and hot plate models. This referee did not find much interesting study and novelty from this manuscript and there are lot of redundancies throughout the manuscript.
Concerns to authors
1. Figure 1, is it synthesis or synthesized compounds?? The authors should provide with details such as how they made and with schematic representation.
2. These are reported compounds in lit.?? If reported authors should include citations as well as NMR spectra should be provided at least some of the compounds for complete characterizations.
3. Typos should be minimized, for example, line 76, 90 space and synthesis of , line 376--Schiff bases and several places space issues were found and should be rechecked again.
4. In line 275 HR-MS spectrometer but I did not see any HRMS in the manuscript.
5. The references styles should be uniform in all cases as per journal guidelines. Ex. Ref 40
Author Response
Reviewer 1
In this manuscript, the authors reported six chiral Schiff bases H1—H3/H4-H6 and these compounds evaluated for analgesic potential at a dose 12.5/25 mg/kg using AcOH induced writhing, formalin test, tail immersion and hot plate models. This referee did not find much interesting study and novelty from this manuscript and there are lot of redundancies throughout the manuscript.
Concerns to authors
Point 1: Figure 1, is it synthesis or synthesized compounds?? The authors should provide with details such as how they made and with schematic representation.
Response: Thank you very much. Worthy reviewer it is synthesis not synthesized compounds. Scheme 1 and 2 has accordingly been added.
Point 2: These are reported compounds in lit.?? If reported authors should include citations as well as NMR spectra should be provided at least some of the compounds for complete characterizations.
Response: These compounds are already known in literature and related citations have been incorporated in the manuscript. The HNMR spectra are attached as supplementary files.
Point 3: Typos should be minimized, for example, line 76, 90 space and synthesis of , line 376--Schiff bases and several places space issues were found and should be rechecked again.
Response: Thank you very much worthy reviewer for highlighting the typos mistake. Action taken and all the typos errors were eliminated from the manuscript.
Point 4: In line 275 HR-MS spectrometer but I did not see any HRMS in the manuscript.
Response: Thank you very much worthy reviewer, the HR-MS has been removed from the line 275 as these analysis were not performed.
Point 5: The references styles should be uniform in all cases as per journal guidelines. Ex. Ref 40
Response: references were accordingly formatted.
Reviewer 2 Report
Please find my comments below:
· The authors' names were written in different fonts.
· The abstract is also written in different fonts, please correct it.
· In the chapter Results and discussion, please describe the performed synthesis, add synthesis schemes with yields. Please remove NMR reports and other spectroscopic data from this section. These data should be included in the Materials and Methods chapter.
· ProszÄ™ dodać w supplementary data wszystkie wymagane widma NMR, zarówno H NMR jak i C NMR, IR i wyniki analizy elementarnej.
· In NMR descriptions of compounds, coupling constants are missing, please complete this.
· Multiplets in HNMR spectra should be written as a range.
· C NMR spectra and elemental analysis of compounds are missing.
· Please standardize the symbol for the melting temperature.
· Please correct the synthesis description in the Materials and Methods section. There are plenty of borrowings from everyday language.
Author Response
Reviewer 2
Please find my comments below:
- Thank you. worthy reviewer, for you positive input. We have tried our best to revise the paper in accordance with your provided instructions. Hope it will be ok now.
- Point 1: The authors' names were written in different fonts.
Response: Thank you, worthy reviewer. They were formatted accordingly.
- Point 2: The abstract is also written in different fonts, please correct it.
Response: Corrected accordingly
- Point 3: In the chapter Results and discussion, please describe the performed synthesis, add synthesis schemes with yields. Please remove NMR reports and other spectroscopic data from this section. These data should be included in the Materials and Methods chapter.
Response: Thank you very much dear reviewer. The synthesis description has been incorporated in form scheme 1 and 2. The spectroscopic data have been moved to materials and method section accordingly.
- ProszÄ™ dodać w supplementary data wszystkie wymagane widma NMR, zarówno H NMR jak i C NMR, IR i wyniki analizy elementarnej.
Response: Worthy editor, after using google translate I understood your point as it is not in English. The information are appended as supplementary file accordingly.
Point 4: In NMR descriptions of compounds, coupling constants are missing, please complete this.
Response: Coupling constant (J values) for all the doublets and triplets in NMR discerptions of the compounds have been added in the revised manuscript.
- Point 5:Multiplets in HNMR spectra should be written as a range.
Response: Multiplets in HNMR spectra have been written as range in the revised manuscript.
- Point 6:C NMR spectra and elemental analysis of compounds are missing.
Response: Thank you, worthy reviewer. We have characterized these compounds only through HNMR and IR spectroscopy as these are known compounds. In case of a known compound if the HNMR peak matches with reported compound, then further advanced spectroscopic analysis are not performed that is common practice. In literature many instances of such studies are there.
- Point 7:Please standardize the symbol for the melting temperature.
Response: Worthy reviewer, symbols were made uniform following international standards. Hopefully they will be ok now.
- Point 8:Please correct the synthesis description in the Materials and Methods section. There are plenty of borrowings from everyday language.
Response: worthy reviewer, the description has been corrected accordingly. Hopefully, it will be ok now.
Round 2
Reviewer 1 Report
The authors responded to some of the worthy comments, still this referee not convinced with the spectroscopic data of some compounds Ex: H3/H5/H6 in SI file and authors should double check again should provide better spectra. Purity is very important for biological aspects, still authors did crystallization and in this aspect data should be reasonable.
Author Response
The authors responded to some of the worthy comments, still this referee not convinced with the spectroscopic data of some compounds Ex: H3/H5/H6 in SI file and authors should double check again should provide better spectra. Purity is very important for biological aspects, still authors did crystallization and in this aspect data should be reasonable.
- Worthy reviewer, thank you for your positive input. The spectral data was corrected accordingly. You are right worthy reviewer purity is very important parameter which is not only determined from crystallization but can also evident from sharp melting point, proton NMR spectra which has been performed in this study.
This manuscript is a resubmission of an earlier submission. The following is a list of the peer review reports and author responses from that submission.
Round 1
Reviewer 1 Report
Title: Synthesis and investigation of analgesic potential of enantiomerically pure Schiff bases: A mechanistic approach
General Comments:
Authors synthesized chiral Schiff bases and evaluated for analgesic potentials at a dose of 12.5 and 25 mg/kg in mice models followed by investigating possible involvement of opioid receptors.
Manuscript needs improvement and this research could be of interest for the field. Please find my suggestions/comments to improve the scientific impact of this manuscript.
- The manuscript needs to be edited and proofread by a colleague with good command of the English language.
- Most of the references in this study are old. Please add newer articles in this field.
- Conclusions to be revised for opioid receptors investigations.
- Line 66; correct ‘to prepared’ to ‘to prepare’.
- Line 210; start the sentence with ‘For’.
- Line 244; Text is not provided under section ‘2. Methodology’.
- Line 293 and 305; change section headings in 4.4.2 and 4.4.3 to italic

Reviewer 2 Report
The manuscript reports on the antinocicpetive characterization of six newly synthesized Schiff bases. The most prominent general impression is, that the authors are mostly experts in chemistry, but they are less familiar with in vivo pharmacological techniques. It is in a strong contrast to the reviewer itself. I as MD pharmacologist can hardly decide whether the chemistry is interesting or not, whether those structures are revolutionary or not. I can decide whether the testing was performed well and maybe the in vivo results are interesting or not. The language of the manuscript is mostly understandable, but sometimes the reader can meet strange sentences (eg. “The extreme analgesic effect of H2 was observed at 90 min latency increased.”?), so a native speaker should reread the manuscript and revise it. Another general finding, that such huge tables with so many numbers are not practical, graphs are mostly much better for presentation such data. It should be changed to graph presentation. If colors are possible, than colored graphs are the best.
Detailed critics:
Methods: Those chapters are mostly inconsistent and extremely superficial.
- Chapter 4.3: Maybe for a chemist those numbers are understandable and clear, if it is a pharmacological publication some description would be necessary. When I understand it, the structure of the compounds was proven by using NMR, but it was not written. It should be rewritten to make it clear for more laic reader as well.
- Chapter 4.4: How were the new compounds solved? Was the solvent saline for all compounds? Possible yes, because saline was used as control, but the authors should indicate it!
- Chapter 4.4.2 What does it men “about 6 mice”. This is a scientific journal, the correct numbers are necessary. On other hand: “0.6 percent of acetic acid”? What was the volume? It is written that mice were divided into 8 groups, but on Fig 2 there are 14 groups?
- Chapter 4.4.3 Number of the animals in different groups are missing. In the description of the formalin test morphine was mentioned as reference compound but in results indomethacin is shown. In the table 1. a “percentage” is mentioned but it is not fully clear what it is. With some logic here it is possible to clear it: it is the % change to control 100%, but the calculation should be shown.
- Chapter 4.4.4 Here it is written, that vehicle is 2% Tween-80. Then what was the solvent? Was it changed between experiments? Here it is written, that latencies were measured after 30 or 15 min, but in the table 2 many time points are included (but not 15 min). Dose of morphine missing. The number of mice / group as well. How was the % effect calculated? There was a cut off time used? Is it MPE%?
- Chapter 4.4.5 The same like in chapter 4.4.4. No group size, no calculation method. Once it is mentioned that jumping or licking was the end sing, but in the next line paw lifting is mentioned as well. What is the truth?
- Chapter 4.4.6 The dose, the injection location (sc. ip. ) and timing of naloxone are missing. Here it is stated that the most potent compounds were tested, but in results section all compounds test are shown.
Result sections
- Chapter 2.1 It contains almost only the informations of Fig. 1. The explanations are mostly in chapter 2.2 (Why?). In chapter there is something about “in-vitro pharmacological investigations”, but there were no such investigations. Chemical investigation (NMR?) is not a pharmacological investigation. A receptor binding assay or some similar would be an in-vitro pharmacology.
- Chapters 2.2: The explanation of results is really confusing. It is very hard to follow the logic of the explanations. And sometimes such sentences are included: “Frequency of writhing was observed and calculated in which the compound H2 showed a significant decrease in the total writhes on all doses.” or “Comparing to the indomethacin a dose of 10mg which 84.55% (26.48±1.33, P<0.05, 127 n=6) and 78.50% (13.21±1.39, P<0.001, n=6).” Interesting result that almost in all cases the “S” counterparts were slightly more potent than the “R”-s but this was neither mentioned nor discussed. The text explains nothing, practically the same numbers are there with some linking text, but nothing else. Other disturbance that in the case of tail-flick and hot plate tests the statistical analysis was made only at 90 min. Here a repeated measure two-way ANOVA would be the most correct method and with post-hoc only the results of the same time point compared. For me another strange thing that morphine had peak at 90 min. It is extremely late. Sc. or ip. morphine has a peak at 30 min and after one hour it strongly declines. In some experiments tramadol was used as well, but why, what was the dose is nowhere mentioned. Because naloxone pretreatment experiments were done fully separated from the first series it is hard to compare the results. For an antagonist testing, agonist only group should have been done at the same time, so this methodology was far not perfect. And therefore no statistic was made to compare the results with or without antagonist.
Introduction and discussion: Both chapters are short. In the introduction there are more paragraphs about pain general with many words about neuropathic pain. But in this project the compound were tested only on acute direct and inflammatory pain models. In the discussion part formalin test mentioned as model for long lasting pain, but this is a typical model for a very intensive short lasting (1h) inflammatory pain and acetic acid writhing is also an acute model. What is really missing from the introduction, is the highlighting why those compounds were constructed in this form, how the authors came to the idea to test them for analgesia? The results show that those compounds (even the most effective H2) are relative weak analgesics compared to morphine or tramadol. The only one model where they have similar effectivity to the reference compound was the formalin test. In the discussion the reader won’t understand more about the pharmacological properties of the compounds since after the simple screening with an exception of partial (central) opioid effect not other analysis was made. Because only two doses were tested it is hard to decide whether this modest effect is the maximum what they can reach or higher doses could be more effective in tail-flick and hot-plate tests as well.
My opinion is that the topic and the work is interesting. The structures could lead to useful medicines once, but the manuscript is extremely badly written, with many mistakes and poor presentation.
Reviewer 3 Report
The present paper describes the synthesis of six chiral Schiff bases and their characterization with particular attetion to the evaluation of analgesic activity.
I have concerns about the publication of the paper.
In my opinion the procedure followed for schiff bases preparation (line 247) “grinding in mortar and pestle for a period of 30 minutes at room temperature” is not standardizable and not reproducible. The force applied during the grinding is not quantified. Tjis parameter is very important because the grinding procedure furnish the energy necessary for the bases formation.
Line 249. Authors state: “The reaction mixture became pasty after grinding. After completion of reaction the products were filtered”. A pasty material cannot be filtered!!!!!!!
The formatting must be reviewed. There are many typing errors.
The english must be revised.
Line 79. It is necessary to indicate a reference supporting the method used to induce pain.
Lines 83, 85, 87, 88: 17.87±2.0 9 unit?
Figure 2. The resolution in very poor.
Lines 237-238. The type and source of reagents used must be indicated.
Line 246: aldehydes with chiral amine. Too generic.
Line 253. Yield. The standard deviation is not reported.
Line 286: weight 18-25gm- gm??
Line 288. Were…..the phrase is not concluded.
Reviewer 4 Report
In this manuscript, the authors report the Synthesis and investigation of analgesic potential of enantio-2 merically pure Schiff bases. The authors have done extensive studies in analyzing these Schiff bases. However, I want to see the rationale of making these specific Schiff bases which I could not find in the manuscript. Please include precedent studies, if any, where Schiff bases were used as analgesics. The manuscript is difficult to follow, the authors should thoroughly go through the manuscript and correct all the typos.